# Alzheimer’s Disease and Age-Related Changes in the Cu Isotopic Composition of Blood Plasma and Brain Tissues of the APP^NL-G-F^ Murine Model Revealed by Multi-Collector ICP-Mass Spectrometry

**DOI:** 10.3390/biology12060857

**Published:** 2023-06-14

**Authors:** Kasper Hobin, Marta Costas-Rodríguez, Elien Van Wonterghem, Roosmarijn E. Vandenbroucke, Frank Vanhaecke

**Affiliations:** 1Atomic & Mass Spectrometry—A&MS Research Unit, Department of Chemistry, Ghent University, 9000 Ghent, Belgium; kasper.hobin@ugent.be (K.H.); martacr@uvigo.gal (M.C.-R.); 2Centro de Investigación Mariña, Universidade de Vigo, Departamento de Química Analítica y Alimentaria, Grupo QA2, 36310 Vigo, Spain; 3Barriers in Inflammation Lab, VIB Center for Inflammation Research, 9000 Ghent, Belgium; elienvw@irc.ugent.be (E.V.W.); roosmarijn.vandenbroucke@irc.vib-ugent.be (R.E.V.); 4Department of Biomedical Molecular Biology, Ghent University, 9000 Ghent, Belgium

**Keywords:** Alzheimer’s disease, APP^NL-G-F^ murine model, blood plasma, brain tissue, Cu isotopes, Cu concentration, ICP-MS/MS, MC-ICP-MS

## Abstract

**Simple Summary:**

Alzheimer’s disease is the most prevalent form of dementia and is associated with multiple alterations in biological processes. The most profound changes consist of the formation of two types of protein aggregations, called β-amyloid plaques and neurofibrillary tangles of tau proteins. Different biologically relevant metals are believed to play a role during the development of Alzheimer’s disease, including copper. In Alzheimer’s disease, copper has been reported to interact with β-amyloid plaques and neurofibrillary tangles, while in addition its homeostasis and metabolism are known to be affected. Therefore, Cu was investigated in a murine model mimicking Alzheimer’s disease. These mice were genetically manipulated, resulting in the presence of typical Alzheimer’s disease symptoms. By making use of two inductively coupled plasma-mass spectrometry (ICP-MS) based analysis techniques, the Cu concentration and the ratio between the natural ^65^Cu and ^63^Cu isotopes (^65^Cu/^63^Cu) were determined in blood plasma and four different brain regions—brain stem, cerebellum, cortex, and hippocampus—of young and aged Alzheimer’s mice and age-matched healthy controls. The Cu concentration in blood plasma was significantly altered in response to both age- and Alzheimer’s-related effects, whereas the blood plasma Cu isotope ratio was affected in the Alzheimer’s-affected mice only. Both the brain stem and cerebellum showed changes in Cu concentration and ^65^Cu/^63^Cu isotope ratio as a result of ageing and the development of Alzheimer’s disease. Finally, a correlation was observed between the ^65^Cu/^63^Cu isotope ratio in the cerebellum and blood plasma. Determination of Cu concentration and isotope ratio is therefore a relevant tool to unravel the role of Cu in ageing and Alzheimer’s disease.

**Abstract:**

Alzheimer’s’ disease (AD) is characterized by the formation of β-amyloid (Aβ) plaques and neurofibrillary tangles of tau protein in the brain. Aβ plaques are formed by the cleavage of the β-amyloid precursor protein (APP). In addition to protein aggregations, the metabolism of the essential mineral element Cu is also altered during the pathogenesis of AD. The concentration and the natural isotopic composition of Cu were investigated in blood plasma and multiple brain regions (brain stem, cerebellum, cortex, and hippocampus) of young (3–4 weeks) and aged (27–30 weeks) APP^NL-G-F^ knock-in mice and wild-type controls to assess potential alterations associated with ageing and AD. Tandem inductively coupled plasma-mass spectrometry (ICP-MS/MS) was used for elemental analysis and multi-collector inductively coupled plasma-mass spectrometry (MC-ICP-MS) for high-precision isotopic analysis. The blood plasma Cu concentration was significantly altered in response to both age- and AD-related effects, whereas the blood plasma Cu isotope ratio was only affected by the development of AD. Changes in the Cu isotopic signature of the cerebellum were significantly correlated with the changes observed in blood plasma. The brain stem showed a significant increase in Cu concentration for both young and aged AD transgenic mice compared with healthy controls, whereas the Cu isotopic signature became lighter as a result of age-related changes. In this work, ICP-MS/MS and MC-ICP-MS provided relevant and complementary information on the potential role of Cu in ageing and AD.

## 1. Introduction

Alzheimer’s disease (AD) is the most prevalent form of dementia and is projected to triple by 2050 [1]. AD is characterized by two major pathological hallmarks, the formation of β-amyloid plaques (Aβ) and neurofibrillary tangles (NFTs) constituted of tau proteins [2]. According to the amyloid cascade hypothesis, Aβ plaques are the first pathological hallmark formed by the cleavage of amyloid precursor protein (APP) by γ-secretases and β-secretases, while α-secretase cleavage of APP prevents Aβ deposition [3]. Aβ segments deposited in the brain can be categorized into two major groups, i.e., Aβ40 and Aβ42. The Aβ42 peptide was shown to be more toxic and has been found responsible for the onset of AD due to the faster growth rate of Aβ42 aggregates [4,5].

Copper is believed to have a relation to the onset of AD due to Aβ’s two Cu-binding sites [6]. Therefore, Aβ plaques formed in the brain of AD patients are known to be enriched in Cu, more specifically in Cu(II) [7]. Copper, as a redox-active element, is able to adopt two different redox states, Cu(I) and Cu(II), which is an ability it shares with only a few other essential elements [8,9]. The redox-active nature of copper allows Cu to initiate the formation of reactive oxygen species (ROS) (H_2_O_2_,·O_2_^−^,·OH) via Haber–Weiss chemistry [6]. ROS generated by Cu-rich Aβ aggregates and unbound Cu in the brain result in an increase in oxidative stress, and the lack of sufficient levels of antioxidants results in cellular damage via oxidation of proteins, DNA, RNA molecules, etc. [10,11].

In the body, absorption of copper from nutrients takes place in the intestinal epithelial cells. Cu(I) and Cu(II) are mainly incorporated by two epithelial membrane transporters, copper transporter 1 and divalent metal transporter-1, respectively [12]. After incorporation and detoxification by several cuproproteins, copper is transported in the bloodstream, predominantly bound to albumin and α-macroglobulin, to the liver [13,14]. In the liver, copper is processed in three different ways: storage within cells bound to metallothioneins, secretion into circulation mostly bound to ceruloplasmin (Cp), and excretion via the bile [13,15].

High-precision isotopic analysis of copper in biological tissues and biofluids, such as plasma, blood, and cerebrospinal fluid (CSF), is capable of revealing natural variations in the Cu isotope ratio related to biological changes [16,17,18,19]. Variations observed in the Cu isotopic composition between different samples and between individuals result from isotope fractionation of the copper isotopes when undergoing biological processes. Fractionation can be driven kinetically, i.e., one of two isotopes experiences a slightly higher reaction rate, or thermodynamically, i.e., the two isotopes show a difference in equilibrium constants. Isotope fractionation occurs upon switching redox state (Cu(I) and Cu(II)) or bonding environment and upon incomplete transfer from one compartment to another. It has been evidenced that the blood plasma Cu isotope ratio is significantly fractionated towards lighter values for patients suffering from Wilson’s disease (WD) [19], end-stage liver disease [18,20], cancer [21,22], and age-related macular degeneration [23] compared with matched healthy individuals. In Parkinson’s disease, changes in Cu metabolism have been revealed via Cu isotopic analysis [24].

The copper concentration and isotope ratio are also being investigated in the context of AD, due to copper being hypothesized to play a role during the pathogenesis of AD [25]. Many works have been published on the association between copper concentration and AD, however, the results remain ambiguous. The Cu plasma concentration of AD patients has been reported to decrease [26,27,28,29], but the opposite trend has also frequently been reported [30,31,32]. Multiple meta-studies investigating the trends observed for the Cu concentrations in blood plasma and serum in the context of AD have been published in the past decade, concluding that serum Cu concentration increases for AD patients compared with controls [32,33,34]. No significant alterations of the copper isotopic composition have been reported yet in murine or human brain and/or plasma samples, but Solovyev et al. [35] reported a trend towards a lighter isotopic composition for brain tissue and plasma in L66 mice (model of frontotemporal dementia) compared with controls. Moynier et al. [36] observed a similar trend towards a lower ^65^Cu/^63^Cu ratio in blood serum and brain tissues of APPswe/PSEN1dE9 transgenic mice (double transgenic mice with both mutations associated with early-onset Alzheimer’s disease). In human brain tissues, the copper isotopic composition was demonstrated to show a trend towards lighter values during AD development [37]. In CSF, Sauzeat et al. [38] did not find a significant difference between AD patients and matched controls, while the CSF Cu from amyotrophic lateral sclerosis (ALS) patients was significantly isotopically heavier than that of controls.

Murine models provide us with the opportunity to assess changes in metal behavior during neurological diseases. To date, the only AD murine models investigated for Cu isotopic composition were one expressing the tau pathology (L66) and two Aβ models (5xFAD and APPswe/PSEN1dE9) [35,36]. In the current work, blood plasma and brain tissues from young and aged male APP^NL-G-F^ knock-in mice and aged-matched wild-type controls were collected for analysis. The APP^NL-G-F^ knock-in murine model affects the levels of pathogenic Aβ via three different mutations, namely the Swedish (NL), the Iberian (F), and the Arctic (G) mutation, increasing the Aβ production, increasing the Aβ42/Aβ40 ratio, and promoting Aβ aggregation, respectively [39]. Brain tissues collected were post-mortem subdivided into four regions, i.e., the brain stem, cerebellum, cortex, and hippocampus, to evaluate spatial differences within the brain. Tandem inductively coupled plasma-mass spectrometry (ICP-MS/MS) and multi-collector inductively coupled plasma-mass spectrometry (MC-ICP-MS) were applied for elemental and isotopic analysis, respectively, to unravel potential age- and/or AD-related changes to the copper metabolism and distribution throughout the body.

## 2. Materials and Methods

### 2.1. Reagents

Ultrapure water (resistivity ≥18.2 MΩ cm) was obtained from a Milli-Q Element water purification system (Millipore, Guyancourt, France). Trace metal analysis grade 14 M HNO_3_ and 12 M HCl were acquired from PrimarPlus (Fisher Chemicals, Loughborough, UK), and a Savillex DST-4000 acid purification system (Savillex Corporation, Minnetonka, MN, USA) was used for further purification. TraceSELECT^®^ 9.8 M H_2_O_2_ was acquired from Sigma-Aldrich (Overijse, Belgium). Isotopic reference material NIST SRM 976 (National Institute of Standards and Technology—NIST, Gaithersburg, MD, USA) was used as an external standard in the Cu isotope ratio measurements. Ga and Cu single-element standard stock solutions (Inorganic Ventures, Nieuwegein, The Netherlands) were used as admixed internal standard and in-house standard solution for quality control of the isotope ratio measurements, respectively. Sample preparation was performed in a class-10 clean lab (PicoTrace, Göttingen, Germany).

### 2.2. Samples

Wild-type (C57Bl/6J) and APP^NL-G-F^ mice (all males) used in this study were housed in the SPF facility at the VIB Center for Inflammation Research. The commercially available APP^NL-G-F^ knock-in mice studied were generated by Saito et al. [39]. The APP^NL-G-F^ mice were bred in C57Bl/6 background and contained the human sequence of exon 16 of the APP gene with the Swedish mutations (KM670/671NL). In addition, the Arctic (E22G) and Beyreuther/Iberian mutations (I716F) were present in exon 17 and a floxed neo cassette was inserted in intron 16 via homologous recombination. These mutations promote Aβ toxicity by increasing total Aβ production (Swedish mutation), increasing the Aβ42/Aβ40 ratio (Iberian mutation), and promoting Aβ aggregation through facilitating oligomerization and reducing proteolytic degradation (Arctic mutation). The mutations result in aggressive amyloidosis, formation of plaques starting at 2 months with near saturation by 7 months, activation of microglia and astrocytes around plaques starting at around 2 months, synaptic loss, and finally, cognitive impairment by 6 months.

After terminal sedation of the mice, blood was collected via cardiac puncture and transferred to EDTA-coated tubes (Sarstedt, Germany). Next, the blood samples were submitted to two steps of centrifugation (10 min, 1300 g, 4 °C; 15 min, 2400 g, 4 °C), after which the supernatant was transferred to pre-cleaned tubes after each centrifugation step. After blood collection, mice were perfused using PBS/heparin, and brain tissues were collected from young (3–4 weeks) wild-type (N = 4), young APP^NL-G-F^ (N = 5), aged (27–30 weeks) wild-type (N = 5), and aged APP^NL-G-F^ transgenic mice (N = 5). Brain tissue was further subdivided into 4 regions: brain stem, cerebellum, cortex, and hippocampus. Samples were immediately stored at −20 °C after collection.

### 2.3. Sample Preparation

The samples were accurately weighed in pre-cleaned Savillex^®^ beakers and 4 mL of 14 M HNO_3_ and 1 mL of 9.8 M H_2_O_2_ were added. The Savillex^®^ beakers were placed on a hotplate at a temperature of 110 °C for 16 h to ensure complete sample mineralization. After digestion, samples were evaporated to dryness at 90 °C and redissolved in 5 mL of 8 M HCl + 0.001% H_2_O_2_.

The chromatographic separation protocol used for Cu isolation from the sample matrices was first described by Maréchal et al. [40]. Resin and elution volumes were fine-tuned by Costas-Rodríquez et al. [18] and Lauwens et al. [41] for low Cu concentration levels and/or small amounts of sample. Chromatographic separation using 1 mL of AG^®^ MP-1M (analytical grade) strong anion exchange resin (chloride anionic form, 100–200 µm dry mesh size, Bio-Rad, Temse, Belgium) loaded into Eichrom polypropylene columns (Eichrom Technologies Europe, Saint Jacques de la Lande, France) was carried out for isolation of Cu from the matrix. Before loading the sample, the resin was pre-cleaned with 10 mL of 7 M HNO_3_, 10 mL of MQ-water, 10 mL of 0.7 M HNO_3,_ and 10 mL of MQ-water and conditioned with 5 mL of 8 M HCl + 0.001% H_2_O_2_. After sample loading, 3 mL of 8 M HCl + 0.001% H_2_O_2_ and 9 mL of 5 M HCl + 0.001% H_2_O_2_ were used to remove the matrix and collect the Cu fraction, respectively. The pure Cu fraction was collected in pre-cleaned Teflon Savillex^®^ beakers and evaporated to dryness on a hotplate at 90 °C. A second identical chromatographic protocol was applied to the Cu fraction to remove residual Na.

Prior to concentration and isotope ratio determination, the collected Cu fraction was evaporated twice to dryness to remove residual chlorides and 500 µL 0.28 M HNO_3_ was used to redissolve the analyte. For every batch of samples (i.e., about 35 samples), the reference material TORT-3 (National Research Council, Ottawa, ON, Canada) was included to evaluate the accuracy of the method and potential loss of the analyte. Loss of Cu during chromatography was evidenced to induce fractionation in the copper fraction. Copper collected in the earlier fractions is known to be fractionated towards a heavier isotopic composition, whereas the subsequent Cu fractions are enriched in the lighter isotope. Quantitative Cu recovery prevents this on-column fractionation from having an effect on the final Cu isotope ratio result. Two blanks following the same procedure were used to evaluate the blank contribution.

### 2.4. Instrumentation and Measurements

An Agilent 8800 tandem ICP-MS/MS instrument (Agilent Technologies, Tokyo, Japan), equipped with a low-flow sample introduction system and Peltier-cooled spray chamber, was employed for Cu quantification. The determination of the Cu concentration relied on external calibration using calibration standards ranging from 0 to 10 µg.L^−1^, with 5 µg.L^−1^ Ga added to all measured solutions (blanks, samples, standards) to correct for instrument instability and/or potential matrix effects. Cu was monitored interference-free as the Cu(NH_3_)_2_^+^ reaction product ion in MS/MS mode via a mass shift from mass 65 to 97; 30% NH_3_ mixed with He was used as reaction gas. Ga was measured “on mass” at mass 71.

Cu isotope ratio measurements were carried out using a Thermo Scientific Neptune *Plus* MC-ICP-MS instrument (Bremen, Germany), equipped with a 100 µL.min^−1^ PFA concentric nebulizer mounted onto a dual spray chamber, comprising a cyclonic and Scott-type sub-unit, and a high-transmission jet interface. Isotope ratio measurements were performed on the interference-free plateau to the left of the spectral peak center, achieved at medium (pseudo) mass resolution. Analyte signals were monitored in static collection mode with all Faraday collectors connected to 10^11^ Ω amplifiers. The instrument settings and data acquisition parameters used are summarized in Table 1 for both instruments. Mass discrimination was corrected for via a combination of internal correction with Ga as internal standard using the approach described by Baxter et al. and external correction with the standard measured in a sample–standard bracketing approach [42,43].

The ^65^Cu/^63^Cu isotope ratio is reported in delta notation, δ^65^Cu (‰) (Equation (1)) relative to the Cu isotopic reference materials NIST SRM 976. The capital delta (∆) is used to denote the difference in ^65^Cu/^63^Cu isotope ratio between two compartments or two groups (Equation (2)).
(1)δ65Cu=Cu65Cu63sampleCu65Cu63NIST SRM 976−1×1000
(2)$${∆}^{65}Cu=\delta^{65}{Cu}_{i}-\delta^{65}{Cu}_{j}$$
where *i* and *j* correspond to different body compartments (i.e., blood plasma and brain tissues) or groups (i.e., AD and healthy).

The δ^65^Cu value obtained for TORT-3 was 0.32 ± 0.06 ‰ (N = 5), which is in agreement with previously reported data [22,44]. The contribution of the procedural blank was 0.5% on average.

### 2.5. Statistical Analysis

IBM^®^ SPSS Statistics 28 (SPSS Inc., Chicago, IL, USA) software for Windows was used for statistical interpretation of the data. Normality of the data was evaluated with the Shapiro–Wilk test. The Kruskal–Wallis and Mann–Whitney U tests were used for non-parametric data to evaluate potential difference between groups. The Spearman rank-order correlation coefficient (Spearman’s Rho, r) was applied for evaluation of the strength and direction of the association between nonparametric variables. A level of significance of *p* ≤ 0.05 was chosen as the threshold value for statistical evaluation.

## 3. Results

### 3.1. Blood Plasma

The copper concentration and isotope ratio results for blood plasma of the APP^NL-G-F^ knock-in mice and the corresponding controls are depicted in Figure 1. The concentrations for Cu in blood plasma samples observed for healthy and AD transgenic mice fell within the range of 69 to 327 µg.L^−1^. A significant increase (Mann–Whitney U, *p* = 0.016) in copper concentration with age was observed in the blood plasma of the healthy controls, however, for AD mice, no increase with age was observed. The blood plasma copper concentration of aged mice showed a significant decrease (Mann–Whitney U, *p* = 0.016) with AD.

Interestingly, the copper isotope ratio in blood plasma provided complementary information to the Cu concentration. Age does not seem to have had an effect on the Cu isotopic signature of the samples, but a trend towards a heavier Cu isotopic composition was revealed for both young (Δ^65^Cu _Y AD–Y Healthy_ = 0.33‰) and aged (Δ^65^Cu _O AD–O Healthy_ = 0.60‰) AD transgenic mice compared with the matched controls. This change towards a higher δ^65^Cu value was significant for the aged mice (Mann–Whitney U, *p* = 0.016), however, significance was not reached for the young mice (Mann–Whitney U, *p* = 0.063).

### 3.2. Copper Alterations in Brain Tissues

The copper concentration and isotope ratio were also determined for the four selected brain regions (brain stem, cerebellum, cortex, and hippocampus) for young and aged AD transgenic mice and matched controls. The results thus obtained are summarized in Table 2. Data are expressed as median and interquartile range (IQR). Concentration values for copper in the brain tissues ranged from 0.86 to 6.23 µg.g^−1^, with the highest copper concentration in the hippocampus, except for aged controls, and the lowest concentration in the brainstem. These results are not in agreement with previous Cu concentrations observed in the different brain areas of an LPS murine model, in which the cerebellum consistently showed the highest Cu concentration for infected and control mice [45].

Age-related changes observed in both the brain stem and cerebellum are presented in Figure 2. The Cu concentration tended to be higher in the cerebellum of aged mice compared with controls, however, this effect was significant for healthy controls only (Mann–Whitney U, *p* = 0.016). No age-related changes were established for the Cu concentration in the other brain regions. Young and aged AD transgenic mice both experienced an enrichment in Cu in the hippocampus compared to healthy individuals, however, without reaching the level of significance. A significant increase in the Cu concentration in the brainstem was observed for both young (Mann–Whitney U, *p* = 0.016) and aged AD transgenic mice (Mann–Whitney U, *p* = 0.008).

The isotopic signature of the brain was heterogenous, with δ^65^Cu values ranging from 0.04 to 0.87‰. The brain stem and the cerebellum showed the heaviest and lightest Cu isotopic composition, respectively. In the brainstem, a significant decrease in the δ^65^Cu value was observed with age for healthy mice (Mann–Whitney U, *p* = 0.016). The cerebellum of both young (Mann–Whitney U, *p* = 0.016) and aged mice (Mann–Whitney U, *p* = 0.016) showed a significant trend towards a heavier Cu isotopic composition for the AD transgenic mice compared with controls.

Figure 3 shows the δ^65^Cu values for the cerebellum plotted as a function of δ^65^Cu values for blood plasma. As can be seen, the cerebellum and blood plasma Cu isotope ratios correlate positively (Spearman’s Rho, r = 0.568, *p* = 0.002). No significant correlations with the blood plasma Cu isotope ratio were observed for the other brain regions. In Figure 3, two distinct groups can be observed: AD transgenic mice revealed higher δ^65^Cu values for both blood plasma and cerebellum, while the controls tended towards lower δ^65^Cu values. Within the control groups, the aged controls showed an even lighter isotopic composition than the young controls.

## 4. Discussion

The copper concentration and δ^65^Cu values obtained for blood plasma and brain tissues were in agreement with values reported for wild-type mice of other murine models in the literature [35,36,45]. Blood plasma Cu concentrations are known to increase with age as a result of increases in inflammation and oxidative stress, leading to a low-grade chronic pro-inflammatory status in the elderly [46,47]. The increase in blood plasma Cu concentrations with age for healthy mice could therefore be related to increased production of the Cu carrier protein in plasma, Cp, in response to elevated cytokine levels, e.g., interleukin-6, interleukin-1β, tumor necrosis factor-α, and interferon-γ [48].

Many publications have reported differences in Cu concentration between AD patients and controls; however, no consistency can be found among these results. As mentioned earlier, Vural et al. [26], Koç et al. [27], and Brewer et al. [28] reported a decrease in blood plasma Cu concentrations for AD patients, while Squitti et al. [30], Siotto et al. [31], and Wang et al. [32] reported an increase instead. In this work, a decrease was observed in the Cu concentration in blood plasma of aged AD transgenic mice compared with controls. For young mice, no significant change with AD was observed. The decrease of the Cu concentration in plasma is in most cases related to a change in the level of ceruloplasmin as the main Cu carrier in blood plasma [49]. For this reason, the decrease observed could be related to a hepatic effect on Cu incorporation, although a change in body distribution of Cu or an increased efflux of Cu cannot be excluded as alternative hypotheses.

The increase in Cu concentration observed in the controls with age and the decrease observed due to AD in the aged mice were accompanied by lower and significantly higher δ^65^Cu values in blood plasma, respectively (Figure 1B). This opposite effect between the Cu concentration and isotopic composition is in agreement with results obtained for a murine model of bile duct ligation, in which the bile duct was sectioned between the duodenum and the hepatic duct and, thus, the bile flow (the major route of Cu excretion) was disrupted. The murine model with a ligated bile duct showed increased Cu concentrations and a significantly lighter whole-body Cu isotopic composition [50]. In WD patients and a murine model for sepsis-associated encephalopathy, an increase of Cu concentration in the blood plasma resulted in a heavier Cu isotopic composition [19,45]. A potential hypothesis for the observed changes in Cu isotopic composition in blood plasma during the pathogenesis of AD is a higher level of Cp without oxidase activity [28]. In order to compensate for this effect, the body may try to achieve a higher saturation of Cp. A more competitive incorporation of the more oxidized form of Cu, Cu(II), in Cp would result in a fractionation of the Cu isotopic composition towards higher δ^65^Cu values in blood plasma [36]. Another hypothesis for the heavier Cu composition and lower Cu concentration observed for aged AD transgenic mice could be related to increased urinary Cu excretion with AD [51]. Cu in urine is believed to be isotopically lighter and heavier Cu would therefore accumulate in the liver, resulting in a heavy Cu isotopic composition of the blood plasma [52].

The Δ^65^Cu value observed between plasma and brain tissues (defined as Δ^65^Cu = δ^65^Cu_plasma_–δ^65^Cu_brain tissue_) is comparable to the difference reported for previously studied murine models; δ^65^Cu values of blood plasma are isotopically 1–1.3‰ lower than those of brain tissue [36,45]. In contrast to Miller et al. [53], who determined the Cu isotope ratio in brain parts of wild-type and prion protein transgenic mice, here the lightest Cu isotopic signature was observed in the cerebellum instead of the cortex, however, similar to our results, the brain stem was enriched in the ^65^Cu isotope.

The most notable effect of AD on Cu concentration was observed in the brain stem, in which both the young and aged mice showed a significant trend towards a higher Cu content. The dysregulated Cu homeostasis in the brain stem of AD transgenic mice could be an indication that the brain stem is affected in an initial stage during the pathogenesis of AD, in contrast to earlier beliefs about the hippocampus that were hypothesized in previous works [54,55]. Although AD does not seem to affect the δ^65^Cu value of the brain stem, age seems to exert a significant effect on the isotopic composition in healthy controls. The isotopic composition becoming lighter as a function of age is not accompanied by a change in the Cu concentration.

The lighter isotopic composition observed in the brain stem of aged controls could be related to a higher fraction of Cu originating from the blood, which is isotopically lighter. The blood–brain barrier and regulation of Cu in the brain are known to be affected by age [56]. Therefore, this trend in the brain stem isotopic composition could hypothetically be attributed to more blood Cu (with a lighter isotopic composition) entering the brain stem. Another possibility is that Cu found in the brains of aged mice is in a stronger bonding environment, preferentially binding the heavier ^65^Cu isotope, or is present in the isotopically heavier oxidized Cu(II) form; e.g., Cu in synapses is mainly found in the Cu(II) form [57].

In the brain, the most profound effect on the Cu isotope ratio was revealed in the cerebellum, where both young and aged mice showed a significant shift towards a heavier Cu isotopic composition in the presence of AD. Whether this trend is related to changes in the cerebellum itself can be questioned due to the correlation observed between the δ^65^Cu values for cerebellum and blood plasma. The latter correlation could indicate a disruption of the Cu balance between the cerebellum and blood plasma, resulting in the Cu signature of the cerebellum being affected by the changes observed in the plasma, or vice versa. When plotting the δ^65^Cu values in cerebellum and plasma versus one another, the AD transgenic and healthy mice can be divided in two clear groups, with a heavier isotopic signature for the AD mice compared to a lighter isotopic signature for the healthy controls. Moreover, ageing in the healthy controls drives the Cu isotopic composition to even lighter values.

In the cerebellum, a significant increase of the Cu content also was observed for the healthy controls with age, which is in agreement with the observation of increasing Cp levels in the brain with age [58]. However, similar to the blood plasma Cu concentrations, changes in the brain Cu content remain ambiguous because the opposite effect has also been reported for both humans and mice [59,60].

## 5. Conclusions

Analysis of wild-type and APP^NL-G-F^ knock-in mice revealed multiple trends in Cu concentration and/or isotopic composition for both blood plasma and brain tissues, due to age- and AD-related changes. Cu concentrations in blood plasma decreased with AD, while AD shifted the isotopic composition towards a heavier signature for aged mice, presumably related to changes in incorporation of Cu in Cp at the hepatic level. The changes in Cu content and Cu isotopic signature in the cerebellum are significantly correlated with those in blood plasma, indicating a potential disruption of Cu homeostasis and a leakier barrier between the cerebellum and the blood plasma. In the brain stem, the Cu concentration showed a significant increase for aged controls compared with young controls, whereas the Cu isotopic signature tended to become heavier for AD transgenic mice.

These results suggest that determination of the Cu concentration and isotope ratio shows potential to become an important tool for the investigation of the mechanisms underlying ageing and development of AD. Combining metallomics, in particular the determination of the copper concentration and the copper isotope ratio, with proteomics, genomics, transcriptomics and/or other complementary analytical techniques could provide a more profound insight into the role of copper and/or copper-related proteins in the pathogenesis of AD.

The heterogeneity of the Cu concentration and δ^65^Cu values measured for the different brain regions emphasizes the importance of the individual analysis of different brain regions, which seem to be affected in different ways. Concentration determination by tandem ICP-MS and isotope ratio determination by MC-ICP-MS were proved to provide complementary information.

## Figures and Tables

**Figure 1 biology-12-00857-f001:**
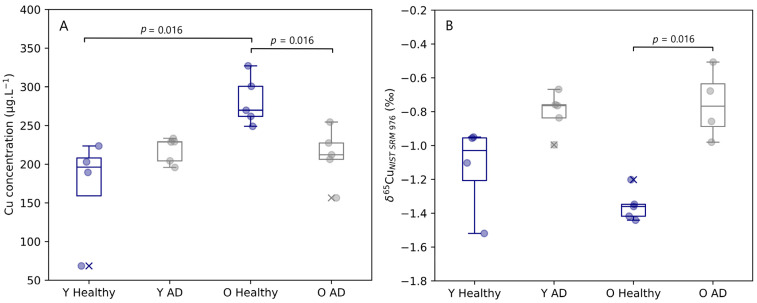
Plasma Cu concentration (**A**) and δ^65^Cu value (**B**) for young and aged APP^NLGF^ knock-in mice and matched controls. Y Healthy, Y AD, O Healthy, and O AD correspond to young controls, young APP^NL-G-F^ transgenic mice, aged controls, and aged APP^NL-G-F^ transgenic mice, respectively. Data for healthy individuals are presented in blue and for AD individuals in grey. Outliers are indicated by a cross. Only significant differences (*p* ≤ 0.05) are indicated in the figure.

**Figure 2 biology-12-00857-f002:**
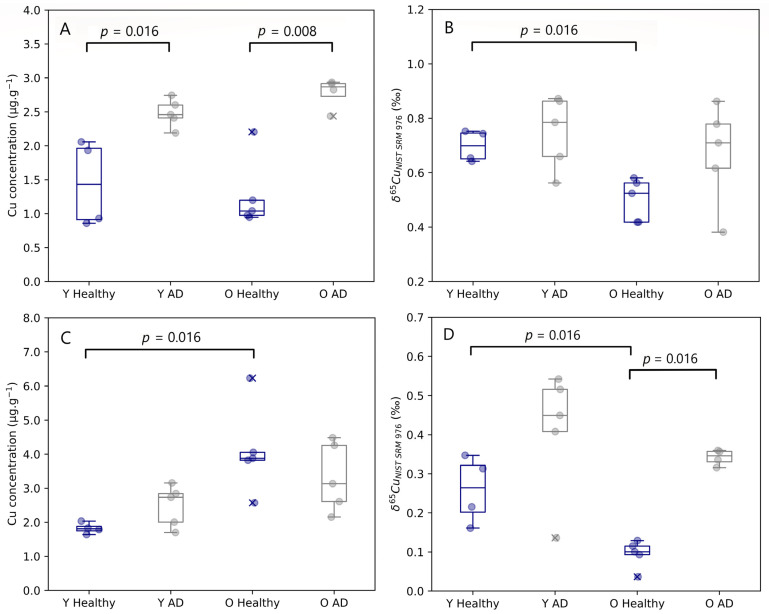
Brainstem Cu concentration (**A**) and δ^65^Cu value (**B**) and cerebellum Cu concentration (**C**) and δ^65^Cu value (**D**) for young and aged APP^NL-G-F^ knock-in mice and matched controls. Y Healthy, Y AD, O Healthy, and O AD correspond to the young controls, young APP^NL-G-F^ transgenic mice, aged controls, and aged APP^NL-G-F^ transgenic mice, respectively. Data for healthy individuals are presented in blue and for AD individuals in grey. Outliers are indicated by a cross. Only *p*-values that indicate a significant difference between populations (*p* ≤ 0.05) are shown.

**Figure 3 biology-12-00857-f003:**
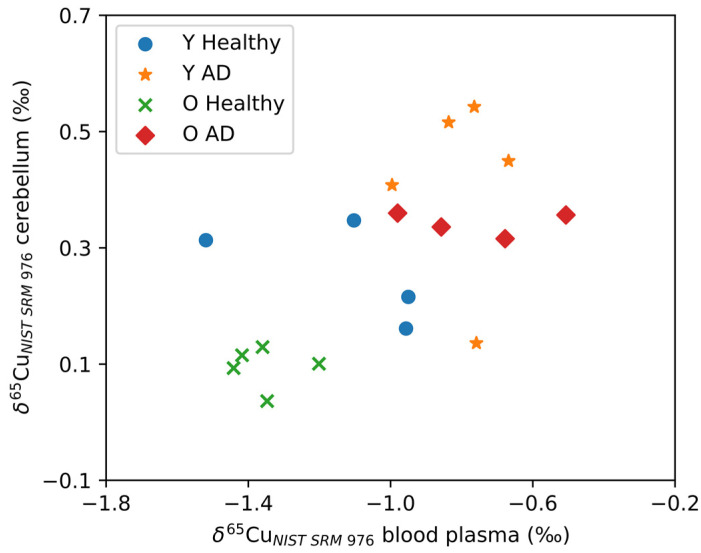
A significant correlation was observed between the δ^65^Cu values for the cerebellum and blood plasma. Y Healthy, Y AD, O Healthy, and O AD correspond to the young controls, young APP^NL-G-F^ transgenic mice, aged controls, and aged APP^NL-G-F^ transgenic mice, respectively.

**Table 1 biology-12-00857-t001:** Instrument setting for the Agilent 8800 ICP-MS/MS and Neptune *Plus* MC-ICP-MS units used for Cu concentration and isotope ratio determination, respectively.

Instrument Settings	Agilent 8800 ICP-MS/MS	Instrument Settings	Neptune *Plus* MC-ICP-MS
Sample uptake rate (µL.min^−1^)	350	Sample uptake rate (µL.min^−1^)	100
Plasma gas flow rate (L.min^−1^)–Ar	15	Plasma gas flow rate (L.min^−1^)–Ar	15
Auxiliary gas flow rate (L.min^−1^)–Ar	0.9	Auxiliary gas flow rate (L.min^−1^)–Ar	0.7-0.8
Nebulizer gas flow rate (L.min^−1^)–Ar	1.12	Nebulizer gas flow rate (L.min^−1^)–Ar	1.0-1.1
Collision gas flow rate (mL.min^−1^)–He	1.0		
Reaction gas flow rate (mL.min^−1^)–NH_3_	3.0		
Rf Power (W)	1550	Rf Power (W)	1200
Sampling cone	Ni tip with Cu base	Sampling cone	Ni, Jet cone
Skimmer cone	Ni	Skimmer cone	Ni, X-type
Data acquisition parameters	Agilent 8800 ICP-MS/MS	Data acquisition Parameters	Neptune *Plus* MC-ICP-MS
Operation mode	NH_3_/He (10%/90%) mode	Scan type	Static, multicollection
Integration time (s)	1	Resolution mode	Pseudo-medium
Replicates	10	Blocks × cycles	9 × 5
Sweeps	100	Integration time (s)	4.194
Nuclides monitored	^65^Cu^+^ → ^65^Cu(^14^N^1^H_3_)_2_^+^, ^71^Ga^+^ (on mass)	Cup configuration—Cu & Ga (IS)	L4: ^63^Cu, L2: ^65^Cu, C: ^67^Zn, H2: ^69^Ga, H4: ^71^Ga

**Table 2 biology-12-00857-t002:** Copper concentration and isotope ratio obtained for the brain stem, cerebellum, cortex, and hippocampus of young and aged AD transgenic mice and controls. Y Healthy, Y AD, O Healthy, and O AD correspond to the young controls, young APP^NL-G-F^ transgenic mice, aged controls, and aged APP^NL-G-F^ transgenic mice, respectively. N represents the number of individuals analyzed.

Groups	Sample Type	Cu Concentration (µg.g^−1^)			δ^65^Cu (‰)		
		Median	IQR	N	Median	IQR	N
Y Healthy	Brain stem	1.54	1.40	4	0.70	0.11	4
	Cerebellum	1.81	0.31	4	0.26	0.16	4
	Cortex	2.88	0.62	4	0.53	0.26	4
	Hippocampus	2.67	1.94	4	0.55	0.21	4
Y AD	Brain stem	2.46	0.37	5	0.78	0.25	5
	Cerebellum	2.74	1.15	5	0.45	0.26	5
	Cortex	3.22	0.63	5	0.63	0.29	5
	Hippocampus	3.51	1.62	5	0.63	0.18	5
O Healthy	Brain stem	1.04	1.00	5	0.52	0.15	5
	Cerebellum	3.88	1.94	5	0.10	0.06	5
	Cortex	3.06	0.69	5	0.44	0.14	5
	Hippocampus	1.54	3.54	5	0.49	0.09	5
O AD	Brain stem	2.91	0.45	5	0.68	0.32	5
	Cerebellum	3.70	2.03	5	0.35	0.04	4
	Cortex	3.22	0.35	5	0.52	0.25	5
	Hippocampus	3.80	1.31	4	0.53	0.22	5

## Data Availability

The raw data supporting the conclusions of this article will be made available by the authors, without undue reservation.

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
