# Peer review of "Alzheimer’s Disease and Age-Related Changes in the Cu Isotopic Composition of Blood Plasma and Brain Tissues of the APPNL-G-F Murine Model Revealed by Multi-Collector ICP-Mass Spectrometry"

_biology, 2023, doi:10.3390/biology12060857_

Round 1

Reviewer 1 Report

The manuscript is of great scientific interest and can contribute to a better understanding of the action of metals in pathological processes such as Alzheimer's disease. The starting hypothesis, which is later confirmed in the results, on the ability of biological systems to selectively bind to the various isotopes of essential metals, such as Cu, and its consequences on their homeostasis and on the AD. The manuscript is methodologically well developed and provides interesting results. However, I am going to introduce a critical consideration, in the sample treatment section. Under my cist point, it would be necessary to justify why the described method has been selected, cite the bibliography on the matter and consider the possible problems associated with potential loss of analytes,

Reviewer 2 Report

In this manuscript, the authors investigated the concentration and the natural isotopic composition of Cu in blood plasma and multiple brain regions (hippocampus, cortex, brain stem and cerebellum) of young and aged Alzheimer’s’ disease (AD) mouse model and wild type controls using tandem inductively coupled plasma-mass spectrometry (ICP-MS/MS). They found multiple trends in Cu concentration and isotopic composition for both blood plasma and brain tissues due to age- and AD-related changes. Overall, the manuscript was well-written and demonstrated again the roles of Cu during the progression of AD. It added another piece of evidence on Cu alterations with the disease and may potentially reveal some underlying mechanisms. Some specific comments are:

1. For Fig. 1 and Fig. 2, it is better to show individual data point in each group as well instead of only box plots.

2. Fig. 3 shows a positive correlation between cerebellum and blood plasma Cu isotope ratios. Is this also the case for the other brain regions or is it specific to cerebellum?

3. As the authors mentioned, the results from different related work are ambiguous and even opposite. It is beneficial to include a more thorough comparison of the current study and the others so the readers can better discern any discrepancies or correlations between them. For example, include a table which lists model system used, analytical method, number of replicates, key observations etc. from each of these available studies.

4. It is also beneficial to include more discussions on what validation methods could be utilized to improve the confidence of the findings, and how different analytical tools are complementary to each other towards a more persuasive study.
